# Evaluation of Selected Operating Process Variables for a Bioflocculant Supported Column Flotation System

**Melody R. Mukandi** [1], **Moses Basitere** [2,*] , **Seteno K. O. Ntwampe** [3], **Mahomet Njoya** [4] , **Boredi S. Chidi** [1] , **Cynthia Dlangamandla** [1] and **Ncumisa Mpongwana** [1]

1    Bioresource Engineering Research Group (BioERG), Department of Chemical Engineering, Faculty of Engineering and the Built Environment, Cape Peninsula University of Technology, P.O. Box 1906, Bellville 7535, South Africa; mukandim@cput.ac.za (M.R.M.); chidib@cput.ac.za (B.S.C.); dlangamandlac@cput.ac.za (C.D.); ncumisam@dut.ac.za (N.M.)

2    Academic Support Programme for Engineering (ASPECT) & Water Research Group, Department of Civil Engineering, University of Cape Town, Private Bag X3, Rondebosch, Cape Town 7700, South Africa

3    Department of Chemical Engineering Technology, Doornfontein Campus, University of Johannesburg, P.O. Box 524, Johannesburg 2006, South Africa; karabosntwampe@gmail.com

4    Atos, 5920 Windhaven Pkwy Suite 120, Plano, TX 75093, USA; mahomet.njoya@gmail.com

*    Correspondence: moses.basitere@uct.ac.za; Tel.: +27-21-650-3238

**Abstract:** The poultry industry generates significant volumes of slaughterhouse wastewater, laden with numerous pollutants, thus requiring pretreatment prior to discharge. However, new technologies must be used to re-engineer the existing wastewater treatment equipment and incorporate new designs to improve the treatment processes or system performance. In this study, three variables, i.e., diffuser design, bioflocculant form, and flow rate, were evaluated to determine their effect on the performance of a bioflocculant-supported column flotation (BioCF) system. It was found that bioflocculants influenced diffuser performance with limited impact when the feed flow rate was varied, i.e., 3D-printed air diffusers and cell-free flocculants imparted high BioCF performance when compared to moulded diffusers and cell-bound flocculants. Notably, the combination of 3D-printed air diffusers and cell-free flocculants resulted in relatively high pollutant removal (81.23% COD, 94.44% TSS, 97.77% protein, and 90.38% turbidity reduction). The study lays a foundation for exploring 3D-printed air diffusers, a relatively new technology in conjunction with microbial flocculants usage that are regarded as eco-friendly for application in industry to enhance the performance of column flotation systems.

**Keywords:** column flotation; bioflocculant form; diffuser design; poultry slaughterhouse wastewater; wastewater





## 1. Introduction

The poultry industry is experiencing significant growth due to the increasing demand for poultry products, which can be attributed to urbanization and burgeoning population growth. However, the industry faces various challenges, with one of the most pressing being the production of large quantities of wastewater due to the high consumption of fresh water [1]. Poultry slaughterhouse wastewater (PSW) is generated during bird slaughter, scald, bleeding, cutting, and packaging, including washing and cleaning equipment and facilities. Disinfectants are also present in the wastewater, leading to the water being classified as polluted [2]. The primary pollutant in PSW is organic matter, including blood, fats, oil and grease (FOG), unprocessed food, and soluble proteins. This results in a high level of CODs and BODs, necessitating the pretreatment of onsite wastewater prior to discharge [3], which has been further complicated by stricter regulations regarding wastewater discharge, as well as the increasing cost of fines [4]. Due to fluctuations in the composition of PSW and influent flow rate, based on the processing stage, the treatment

method must be adapted to cater to these variations [5]. However, the inherent quantity and quality of PSW from the poultry industry requires the improvement of the treatment processes. Therefore, developing new technologies, re-engineering existing wastewater treatment equipment, and incorporating new designs are essential to enhance the treatment processes or system performance [6].

PSW has been treated with a variety of techniques, including flotation. The mineral-processing industry gave rise to this gravity separation technique, which is today extensively used in water and wastewater treatment. Industrial wastewater is often laden with contaminants and must be remediated before discharge or used for any other purpose [7]. Flotation has been used in wastewater treatment to remove difficult-to-separate particles, FOG, and residual compounds [8,9].

On the other hand, flotation is a separation process whereby particles with a low density either float to the top of the medium or settle at the bottom and are separated. This is facilitated by air bubbles forming bubble-particle aggregates that rise to the top, where they are subsequently scraped off [8]. This implies that the flotation process is based on the particles' attachment to air bubbles. Bubbles serve as a transport medium for flotation particles. Hence, bubble creation or generators are an important component of a flotation system. Bubbles are primarily generated through two broad categories, i.e., dispersed and dissolved air [7]. In dispersed air, bubbles are generated by directly supplying compressed air into bubble generation devices, including microporous tubes, diffusion discs, and hydraulic injectors. In contrast, dissolved air involves dissolving air in water under pressure and then releasing it at atmospheric pressure. The generation cost for dissolved air is higher than that of dispersed as it requires high energy and rigid operating conditions. Hence, there is a need for an easy operation development [10].

Various forms of flotation are used in separation processes, including induced air flotation (IAF), dissolved air flotation (DAF), Jet, Ion flotation, and column flotation. Notably, DAF has superior separation efficiency in wastewater treatment but is expensive due to high water saturation costs [9]. For this study, column flotation was explored as it is relatively cheap. The advantages of a column flotation are higher separation efficiency, lower capital required due to the simplicity of the system setup, and lower operating cost due to the lack of complexity of the process [11].

A column flotation is a vessel that is at least twice as tall as it is wide. It is a separation process driven by variations in surface properties, and air-sparging devices are employed to generate the necessary air bubbles that enable separation and removal. Air bubbles are introduced near the bottom, and these bubbles adhere to chemically altered and naturally hydrophobic particles, forming bubble-particle aggregates [12,13]. The flocculation step, which involves flocculants, is employed to improve the process efficiency for bubble particle capture. Flocculation has long been widely used to remove pollutants in wastewater and water treatment due to its effectiveness and convenience [14]. Flocculants are used to aid particle aggregation and facilitate floc formation as they act as bridging compounds. This is typically due to ionic and hydrogen bond formation and electrostatic interactions [14,15]. According to green chemistry, chemical flocculants are disadvantageous as they produce toxic sludge and are non-biodegradable. Moreover, bioflocculants are considered safe to handle, biodegradable, and environmentally benign. This is why bioflocculants are becoming increasingly popular [16]. Bioflocculants of microbial origin are basically non-toxic extra polymeric substances formed by the microbe or its metabolites, which include proteins, polysaccharides, DNA, glycoproteins, and others. In other words, bioflocculants are macromolecules and these molecules contain hydrophilic groups such as carboxyl and hydroxyl groups that favour floc formation by adsorption bridging [17]. Bioflocculants originate from lysis or the metabolism of microorganisms. They are subdivided into soluble and bound flocculants, with bound being attached to the cell whereas soluble flocculants are dissolved in a solution or are weakly bound with cells [18]. These forms can be separated by centrifugation [19]. However, information regarding these two types of bioflocculants is limited [20]. Bioflocculants of microbial origin will be employed in this investigation.

Various factors, including operating variables, equipment, and chemicals, influence flotation processes. This, in turn, influences the removal efficiency [21]. Despite being developed in the 20th century, it still needs to be better understood and could be more efficient [22]. As research on the mechanism and processes of flotation has intensified, its drawbacks have been observed in high cost, large equipment, intricate processes, and bubble sizes, which can lead to variations in removal efficiencies [10]. Furthermore, studies conducted in the past have indicated that aeration devices may only sometimes be optimized [12], and recent attention has been paid to new technologies that enhance aeration processes [23] and, in this case, 3D printing will be considered.

Three-dimensional printing, also known as additive manufacturing or rapid prototyping, produces an object through layer-by-layer fabrication using computer-aided design drawings (CADs). Three-dimensional printing offers flexibility in terms of design specifications. The technology can produce an object using a wide range of materials including hybrid combinations [24]. Materials that include polymers, pure metals, metal alloys, composites, ceramics, and thermoplastics can be used in 3D printing [25]. The primary difference between 3D printing and traditional methods is that the former involves an additive approach producing minimal to no waste whilst the latter involves a subtractive approach, meaning additional waste is produced [26,27]. The conventional/traditional techniques include a combination of bending, moulding, cutting, gluing, welding grinding, and assembling. Three-dimensional printing has led to the reduction in build time as compared to some of the traditional manufacturing techniques, which take longer due to the steps/processes involved [26]. Another benefit of 3D printing is that it fosters innovation as it has the ability to print using a wide range of materials with limited restrictions in the production of complex structures as compared to the traditional methods in which other materials cannot be utilized [27,28]. Another distinction is that 3D printing has a competitive advantage due to its ease of customisation and also ease of manufacturing geometrically complex parts [29], e.g., irregular shapes, variable thickness, hollow interiors, etc., which can be produced based on CAD. This leads to multimaterial, lightweight, ergonomic products, etc. Though 3D printing has advantages over traditional methods, it will not replace them completely, but rather revolutionize the production methods [30].

The production or use of 3D-printed air spargers has yet to be explored. Hence, the current research focused on the design of diffusers as a parameter for comparing 3D-printed and moulded diffusers. Additionally, bioflocculation forms and feed flow rates were also considered as key parameters to assess their impact on overall system performance for a bioflocculation-supported column flotation for pretreatment of PSW. The study lays a foundation for the exploration of 3D-printed air diffusers, a relatively new technology in conjunction with microbial flocculants for application in industry to enhance the performance of column flotation systems.

## 2. Materials and Methods

### 2.1. Column Air Flotation Bench Scale Setup

This study employed a column flotation tank similar to the one previously designed by [31] (2016), albeit with minor modifications. Figures 1 and 2 depict the schematic and photographic illustrations of the column flotation tank. The column flotation tank's design featured a cross-sectional shape to maximize the surface area, an inlet near the top, and one sampling point (outlet) positioned just below the inlet but on the opposite side of the column. This configuration facilitated the separation of formed flocs from PSW and the pretreated wastewater. The components of the column flotation system were connected using silicon tubing. PSW from a 2 L holding tank was continuously fed into the plexiglass column flotation tank with an adequate volume capacity of 1.13 L via a Gilson peristaltic pump, with the flow rate being varied based on response surface methodology (RSM). *B. megaterium*-derived D2 flocculants were utilized for the flocculation process. Compressed air, regulated by pressure gauges and an airflow meter, was injected into the column flotation tank through air diffusers, resulting in a bubbling stream. Microbubbles

produced at the tank's base attach to the flocs/solids, forming bubble-particle aggregates that rise to the top of the tank, where they are subsequently skimmed. Samples were collected at predetermined intervals and were analysed for quality water parameters. All experiments and tests were conducted at room temperature.

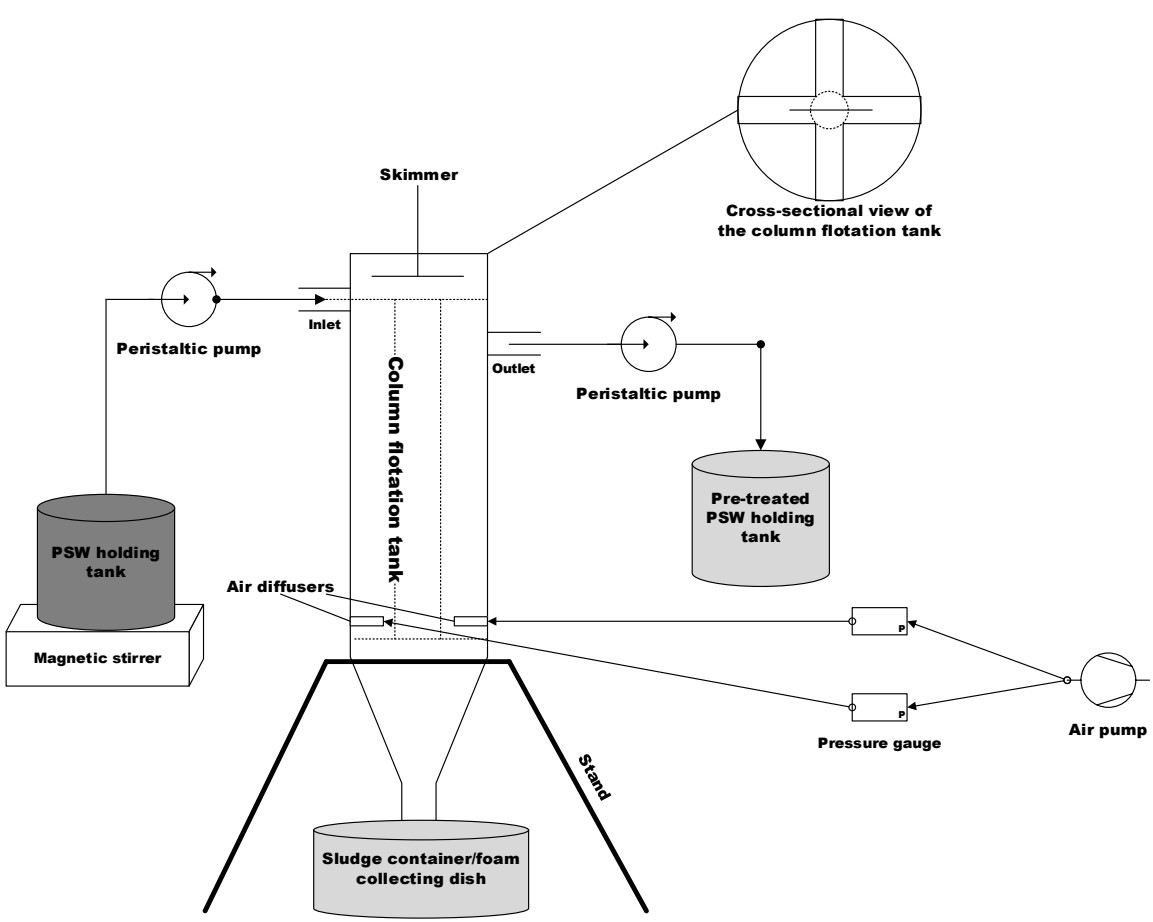

**Figure 1.** Schematic illustration of a column flotation system setup.

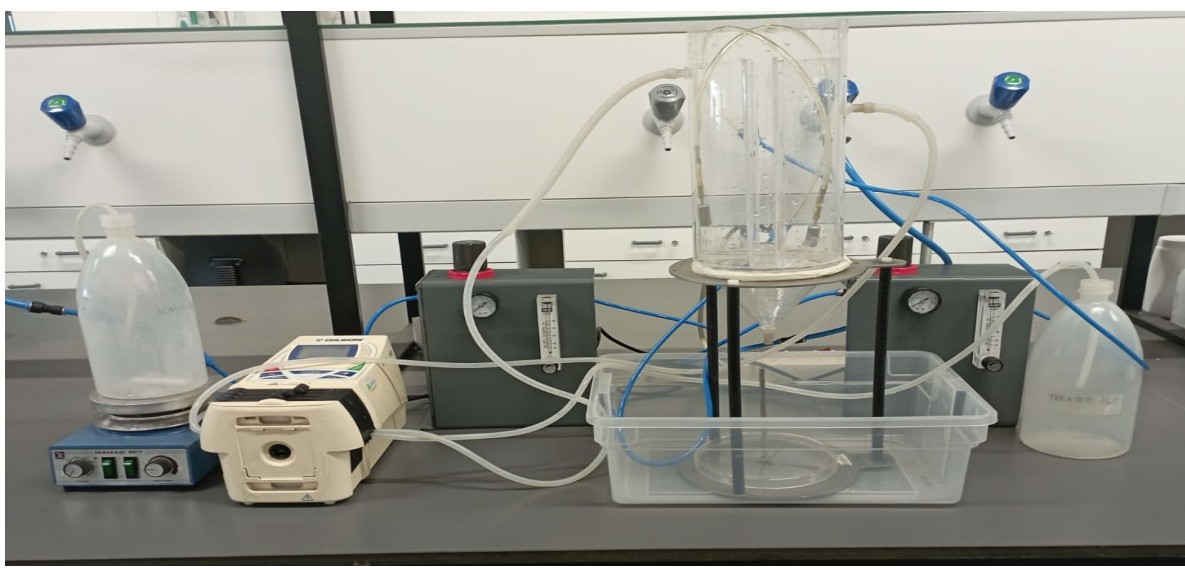

**Figure 2.** Photographic illustration of a column flotation.

*2.2. Wastewater Source*

PSW was obtained from a poultry slaughterhouse in Cape Town, Western Cape, South Africa. The water was explicitly collected from the slaughtering plant, whereby PSW generated was from the slaughtering and washing of birds, cleaning of surfaces, and processing of by-products [32]. The wastewater was collected in 25 L polypropylene containers and stored at 4 °C before use to inhibit/lessen any biological activity. The PSW characteristics were analysed using standard methods prior to and post-pretreatment. Table 1 lists the average initial PSW parameters before running the system.

**Table 1.** Average PSW parameters prior to pretreatment.

| Parameter | Average |
|---|---|
| pH | 6.64 |
| COD | 2017.62 mg/L |
| Turbidity | 449.87 NTU |
| Suspended solids | 836.15 mg/L |
| Protein | 370.68 µg/mL |

*2.3. Bioflocculant Production and Flocculation Activity Confirmation*

*B. Megaterium*, previously isolated from PSW, was used for bioflocculant production. A loopful of the bacteria from nutrient agar plates was transferred into 50 mL bioflocculant production media as formulated by [33] (2017), and was incubated in a shaker incubator (Labwit ZWYR-240 shaking incubator, Labwit Scientific, Burwood East, VIC, Australia) at 36.5 °C under 121 rpm for 24 h. Following the incubation period, 5 mL of the fermentation broth was further transferred into 45 mL of bioflocculant production media and was incubated under the same conditions as the inoculum. The resultant fermentation culture broth was used for cell-bound bioflocculants (as is) and cell-free bioflocculants (supernatant after centrifugation to remove cells). The flocculation activity was then quantified using 4 g/L kaolin clay suspension whereby 50 mL of the suspension was aliquoted into 250 mL conical flasks and 1.5 mL of $CaCl_2$ (1% $w/v$) and 1 mL of the bioflocculant for sample or 1 mL of distilled water for control were added to the suspension. The mixture was swirled and transferred into 50 mL measuring cylinders where it was allowed to settle for 5 min. A sample was then withdrawn from the top layer and its optical density was read at 550 nm using a spectrophotometer (Jenway 7305 Spectrophotometer, Bibby Scientific Ltd., Staffordshire, UK). The flocculation activity was calculated using Equation (1), with the quantification being carried out in duplicate, and this served as confirmation that indeed the bioflocculants were effective.

$$\%Flocculation\ Activity = \frac{A - B}{A} \times 100 \tag{1}$$

where:

$A$ = absorbance of control, and
$B$ = absorbance of sample.

*2.4. Using Design Expert Software for Performance Analysis of the Bioflocculant Column Flotation System*

Response surface methodology in Design Expert is an assemblage of statistical and mathematical tools used in process optimization. Furthermore, it can be used to evaluate the significance of multiple parameters in intricate interactions [34], and it was used to optimize the input parameters. The effect of different operating parameters in removing pollutants (COD, TSS, turbidity, and protein) was assessed for experimental design. This involved varying the influent flow rate, use of different bioflocculant forms (including those with and without cells), and utilizing different diffusers (moulded vs. 3D printed diffusers), which had an effect on bubble size and formation, thus ultimately having an

impact on pollutant removal. The range of the flow rate values used was based on the trial run. Chemical oxygen demand reduction, suspended solids removal, turbidity, and protein reduction rate were analysed as responses ($Y$), and the yields were calculated as a percentage removal using Equation (2).

$$Y_n\% = \frac{Y_a - Y_b}{Y_a} \times 100 \tag{2}$$

where:

- $Y_n\%$ is the yield (COD or tSS or turbidity) percentage removal,
- $Y_a$ is the response variable initial value, and
- $Y_b$ is the response variable final value.

Randomized optimal design (custom) was used as there was one numerical factor and two categorical factors. However, based on the design matrix, 18 experimental runs were conducted. Table 2 presents the randomly optimized (custom) 18 generated runs for one numerical factor and two categorical factors, the conditions used in this study.

**Table 2.** Experimental design for the independent variables using central composite design matrix.

| | Factor 1 | Factor 2 | Factor 3 |
|---|---|---|---|
| **Run** | **A: Feed Flow Rate (mL/min)** | **B: Diffuser (Type)** | **C: Bioflocculants (Appendage Type)** |
| 1 | 1.26 | Moulded | Cell bound |
| 2 | 1.37 | 3D printed | Cell bound |
| 3 | 1.26 | 3D printed | Cell free |
| 4 | 1.00 | 3D printed | Cell bound |
| 5 | 1.00 | Moulded | Cell bound |
| 6 | 1.00 | Moulded | Cell free |
| 7 | 1.00 | 3D printed | Cell free |
| 8 | 1.74 | Moulded | Cell free |
| 9 | 2.00 | Moulded | Cell bound |
| 10 | 1.74 | Moulded | Cell free |
| 11 | 2.00 | 3D printed | Cell free |
| 12 | 1.73 | 3D printed | Cell bound |
| 13 | 2.00 | 3D printed | Cell free |
| 14 | 1.37 | Moulded | Cell free |
| 15 | 1.26 | Moulded | Cell bound |
| 16 | 2.00 | 3D printed | Cell bound |
| 17 | 2.00 | Moulded | Cell bound |
| 18 | 1.26 | 3D printed | Cell free |

*2.5. Analytical Methods*

PSW samples collected before and after pretreatment using a column flotation system were tested for water quality parameters. pH, COD, TSS, turbidity, and protein concentration were analysed. The pH of the wastewater was measured using a pH meter (Crison PH 25 plus, Crison Instruments s.a., Barcelona, Spain). Turbidity measurements were performed using a portable TURB 355 IR turbidimeter. SS concentration was determined according to the EPA Method 160.2. COD was determined according to the U.S. Environmental Protection Agency (EPA) 410.4 procedure for COD determination of surface and wastewater [33]. This involved utilizing HANNA high range (HI93754C-25 HR, 0–15,000 mg/L) COD test kits. Protein concentration was determined using the Bradford Assay (BIO-RAD Quick Start™ Bradford protein assay kit, Hercules, CA, USA), and the correlation coefficient was used to check the accuracy of the line obtained from plotting the absorbance against the known concentrations of a protein.

*2.6. Statistical Analysis*

Statistical analysis was performed using Microsoft Excel, Origin 2018 graphing, and Python libraries such as Pandas, Matplotlib, Scipy, and Seaborn. Furthermore, the results were presented regarding the average of at least duplicates.

## 3. Results

It was previously highlighted that equipment, various chemicals, and other operational parameters [21] influence flotation processes. Therefore, this study's selected operating process variables were sparger design, bioflocculant form, and feed flow rate. Design Expert version 11 generated the conditions through RSM. This statistical and mathematical approach allows the simultaneous analysis of multiple factors at various levels, including their combined effect on the response [35].

*3.1. Influence of Bioflocculants on Poultry Slaughterhouse Wastewater Pretreatment*

Bioflocculants, used in water and wastewater treatment, destabilize particulate matter and act as a bridging agent. This leads to the aggregation of particles, thus forming flocs, which are easily separable [36]. The flocculation efficacy of the bioflocculant D2 was first validated using kaolin suspension; therefore, its potential in the treatment of PSW was evaluated by analysing the TSS before and immediately after adding bioflocculants at time 0. The impact is illustrated in Figure 3.

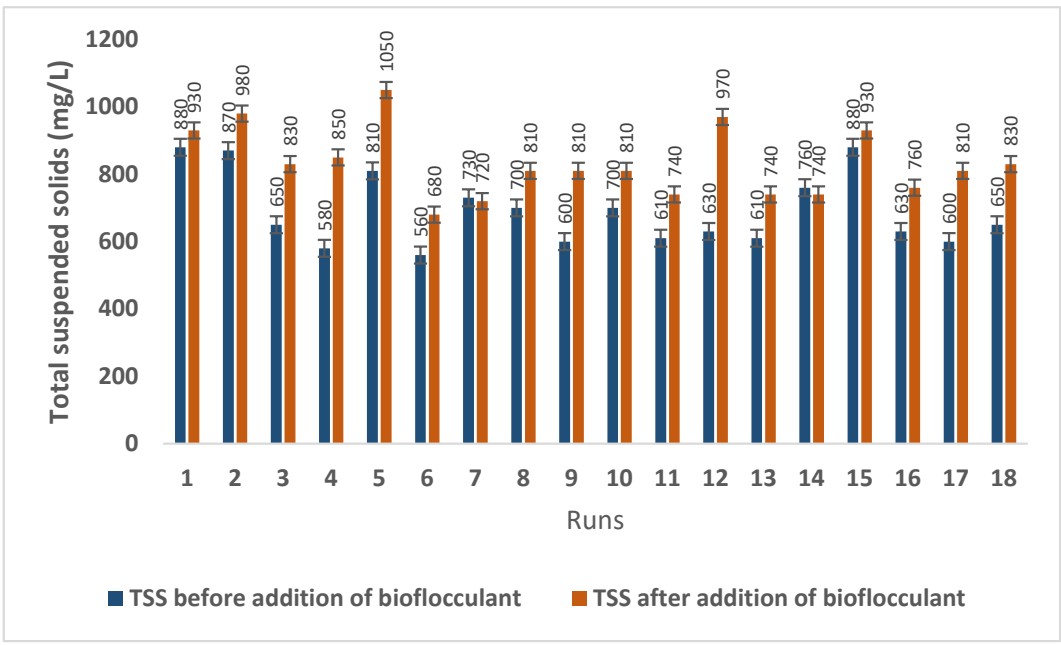

**Figure 3.** TSS before and after the addition of bioflocculants with run 3, 6, 7, 8, 10, 11, 13, 14, and 18 having cell-free bioflocculants and run 1, 2, 4, 5, 9, 12, 15, 16, and 17 having bioflocculants with cells.

The effect of different forms of flocculants, i.e., between cell-free compared to bound flocculants, was exhibited by the variation in TSS reduction. The results also demonstrated that supplementing flocculants to the PSW increases the turbidity of the resultant PSW-bioflocculant water. This distinction could also be attributed to the aggregation of particles, thus resulting in the formation of flocs, as evidenced in the withdrawn samples. The results concur with those of [37] (2023), which confirmed the flocculation of particulate matter in natural water apart from kaolin clay suspensions. Its results indicated that the suspended solids were flocculated efficiently when the flocculants were added alone. Interestingly, its bioflocculant was from Bacillus sp., although the strain was not mentioned.

Furthermore, run numbers 4, 5, 9, 12, and 17 significantly changed after adding cell-bound bioflocculants, resulting in increments for TSS between 210 and 340 mg/L, i.e., the difference between before addition of bioflocculants and after the addition of bioflocculants. In contrast, runs supplemented with cell-free bioflocculants had a TSS range between 110 and 180 mg/L, i.e., the difference between before and after addition of bioflocculants. This difference was initially attributed to bacterial cells. Additionally, flocculants convert dissolved solids/soluble matter into tiny particles that form insoluble complexes and become part of the flocs [38], contributing to the increase in TSS. The findings suggest that supplementing bioflocculant D2 positively affected the PSW and subsequently impacted the flotation system's performance by increasing pollutant removal efficiency. This underscores their potential applicability as a substitute for chemical flocculants.

### 3.2. Comparative Analysis of Comparable Variables

A comparative analysis was conducted to determine the effect of the bioflocculant form (cell free vs. cell bound) and diffuser type (3D printed vs. moulded) to find a combination that performs better. This was accomplished by analysing results when the system was operated under the same operational conditions. Figure 4 illustrates pollutant removal efficiency.

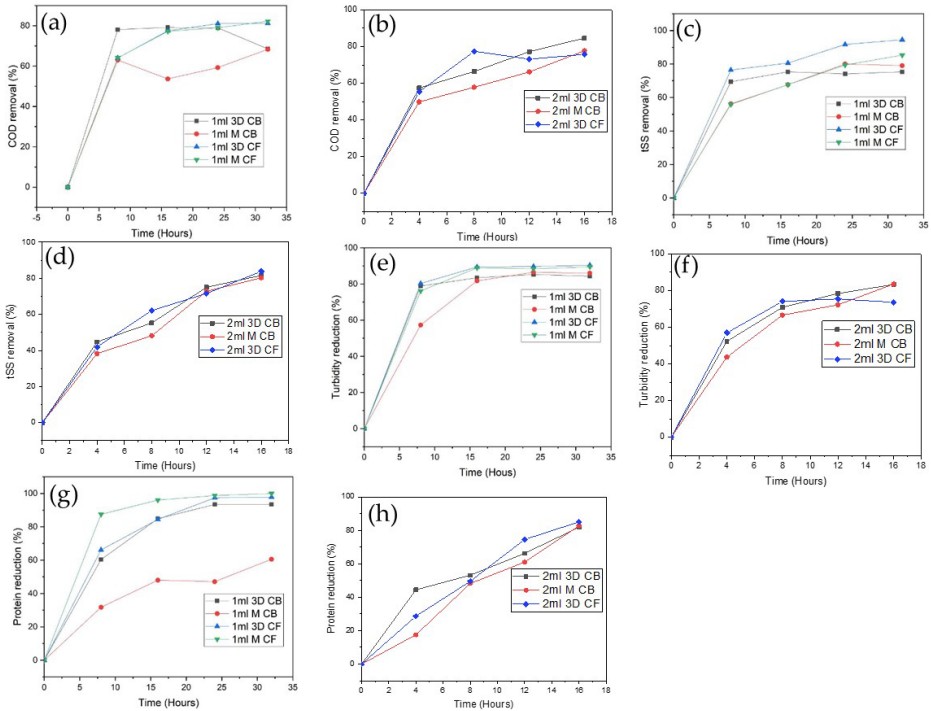

**Figure 4.** Graphical representation of pollutant removal, i.e., COD (**a**,**b**), TSS (**c**,**d**), turbidity (**e**,**f**), and protein under specific operational conditions (**g**,**h**). 3D = 3D-printed diffusers, M = Moulded diffusers, CB = Cell-bound bioflocculants, CF = Cell-free bioflocculants, 1 mL = 1 mL/min and 2 mL = 2 mL/min.

### 3.2.1. Comparison of Bioflocculant Forms

The effect of the bioflocculant form (cell-free vs. cell-bound bioflocculant) was evaluated concerning pollutant removal while maintaining a consistent inflow rate and diffuser type. Bioflocculants generally exist in two forms, i.e., cell-bound bioflocculants, which are affixed and within bacterial cells, and soluble bioflocculants, which are dissolved in a solution as extracellular by-products [39]. Crude bioflocculants obtained after centrifugation were used as cell-free flocculants, and the fermentation broth without centrifugation, which had both soluble and bound bioflocculants, was used as cell-bound flocculants. Figure 4a,c,e,g shows that when 3D-printed diffusers when used at a feed flow rate of

1 mL/min with either cell-free or cell-bound flocculants, the pollutant removal was high for the cell-free flocculation system compared to when cell-bound flocculants were used. Similarly, when moulded diffusers are employed at a similar 1 mL/min inflow rate, the cell-free flocculation system had higher pollutant removal than the cell-bound flocculant system.

Overall, the trend demonstrated that cell-bound flocculants had an inadequate pollutant removal rate, with protein reduction at its lowest at only 60%. However, increasing the inflow rate when using 3D-printed diffusers to 2 mL/min showed that cell-bound flocculants were more effective in reducing COD and turbidity than when cell-free flocculants were used. On the other hand, the reduction in proteins and TSS was only marginally more significant for cell-free flocculants.

Based on our findings, it was deduced that cell-free flocculants were superior to cell-bound flocculants because they yield better overall pollutant removal efficiencies. The lower removal efficiencies with cell-bound flocculants may be attributed to the proliferation of microorganisms during flocculation or the effect of other constituents used in the broth. Non-settleable microorganism growth increases turbidity and reduces pollutant removal [40], as bacterial cellular membrane functional groups can bind some flocculants, reducing flocculation activity [41].

### 3.2.2. Comparison of Diffuser Types

The effect of diffuser design was assessed, with pollutant removal being the outcome. The difference in the microporous structure is mainly dependent on the fabrication method. The 3D-printed air diffusers were manufactured using the laser-powder bed fusion technology, while the diffusers used for comparative analysis were fabricated using the traditional method of moulding/sintering. Pollutant removal for 3D-printed diffusers, when compared to moulded diffusers under similar operational parameters (flow rate and bioflocculant form), is displayed in Figure 4a–h.

Three-dimensional printed air diffusers had better performance when compared to the moulded variety. While COD removal and turbidity reduction were roughly similar for both diffuser types, TSS removal was higher for 3D-printed air diffusers. However, protein reduction was slightly higher for moulded diffusers using cell-free flocculants at 1 mL/min (Figure 4a,c,e,g). However, under the same circumstances, the moulded diffusers' TSS removal and turbidity reduction were marginally more significant than those of 3D-printed diffusers. A closer examination reveals that 3D-printed air diffusers had high removal efficiencies of above 70% just after the beginning of the experiments and continued to rise till the end at operating parameters of 1 mL/min with cell-bound bioflocculants, whereas the moulded diffuser had 60% removal rates, which dropped and subsequently rose although below that of the 3D-printed diffuser experiments. At 2 mL/min with cell-bound bioflocculant (Figure 4b,d,f,h), COD removal was higher for 3D-printed diffusers than for moulded diffusers. However, TSS removal, protein, and turbidity reduction at the end were more or less the same. However, it is noteworthy that the trend indicates that the performance of 3D-printed diffusers started higher than that of moulded diffusers.

These results demonstrate that the type of diffusers affected the performance of the flotation system, with 3D-printed air diffusers outperforming moulded ones in terms of performance. Three-dimensional printed air diffusers had a rough finish compared to the moulded ones, which had a smooth finish. With that said, the surface finish of the 3D-printed air diffusers might have contributed to keeping bubble sizes favourable for flocculation by preventing bubbles from coalescing. This, however, supports the idea that the 3D-printing of diffusers has the potential to enhance column air flocculation system performance. Hence, further exploration is needed to determine the effect of 3D-printed diffusers with dense pores when used for flocculation.

### 3.3. Correlation of the Variables

Pearson correlation coefficient serves as a measure of the linear association between two sets of variables. To investigate the relationship between the physicochemical pa-

rameters, that is, the pollutant removal efficiencies based on operational parameters, the assessment of correlation coefficient was applied. The matrices presented in Figure 5 showed that the correlation coefficients were generally very high, indicating a strong correlation between the physicochemical parameters. Upon closer inspection, turbidity and TSS strongly correlated for systems in which 3D-printed diffusers and cell-free bioflocculant were used (Figures 3b and 3c, respectively). The least, albeit still high, correlation was between turbidity and protein removal at 0.78 (Figure 5a).

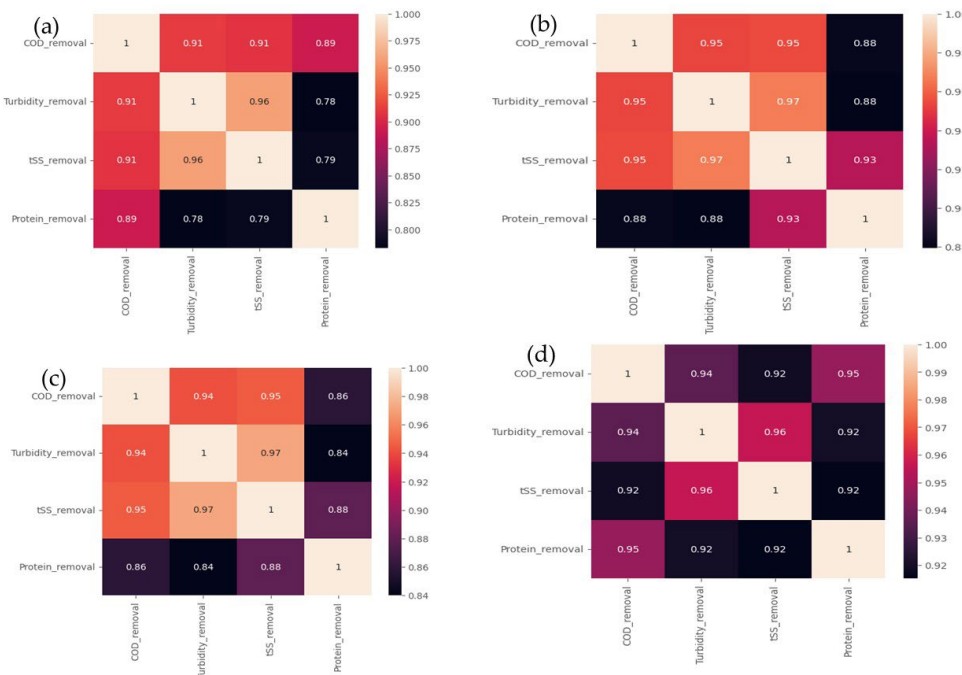

**Figure 5.** Pearson correlation coefficient of different physiochemical parameters based on different variables showing correlation matrix of removal percentages for moulded diffusers (**a**), 3D-printed diffusers (**b**), cell-bound bioflocculants (**c**), and cell-free bioflocculants (**d**).

Regarding all variables assessed, 3D-printed air diffusers and cell-free flocculants constitute an amenable combination that may improve column air flocculation system performance on a large scale.

### 3.4. Surface Plots of the Comparable Variables Based on Pollutant Removal

The surface plots show pollutant removals for various variables based on the flow rate and time interaction. The pollutant removals for cell-bound bioflocculant, cell-free bioflocculant, moulded diffusers, and 3D-printed air diffusers are displayed in Figures 6–9.

For cell-bound bioflocculants (Figure 6), the high pollutant removal rates of between 70% and 80% were attained within 10 to 15 h of operating the system except for protein removal, which extended between 15 h and 20 h. Overall, protein removal was unsatisfactory, as indicated by hue, which denotes low removal efficiencies. Furthermore, the plots demonstrated that turbidity removal was relatively high for the range of flow rates studied.

The surface plots for cell-free bioflocculants (Figure 7) indicate that the protein removal was comparatively more significant than that of cell-bound bioflocculants. There was high COD, turbidity, and protein removal from 5 h, with TSS removal occurring after 10 h.

High removal rates were seen in the Figure 8 plots for moulded diffusers after 10 to 15 h of system operation. However, protein removal fluctuated and had significantly low removals at nearly 20 h.

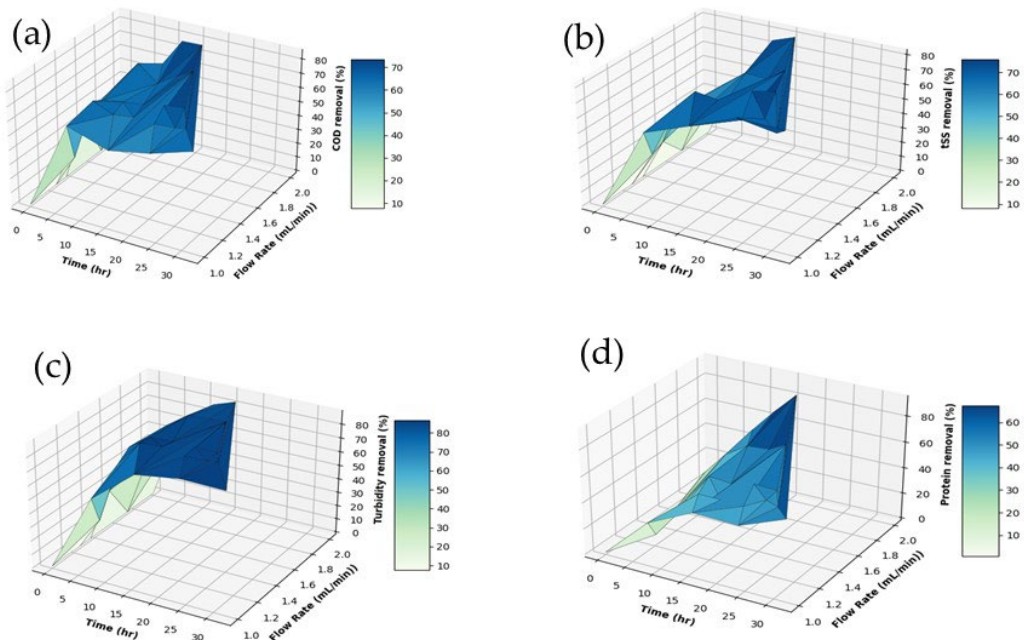

**Figure 6.** Surface plots of (**a**) COD, (**b**) TSS, (**c**) turbidity, and (**d**) protein removal from cell-bound bioflocculants.

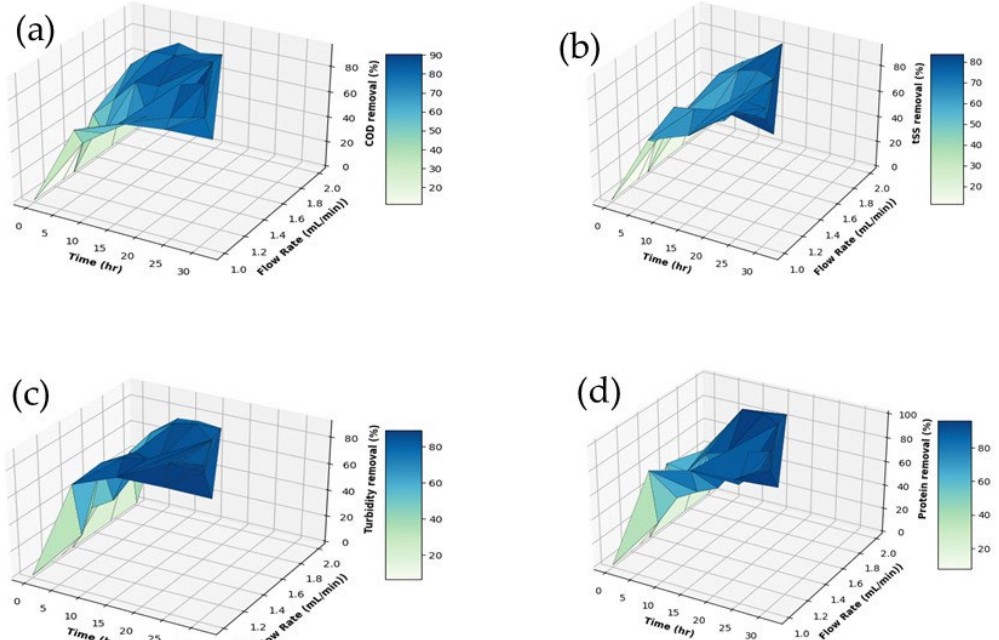

**Figure 7.** Surface plots of (**a**) COD, (**b**) TSS, (**c**) turbidity, and (**d**) protein removal from cell-free bioflocculants.

In line with the findings of other variables, the surface plots (Figure 9) for 3D-printed air diffusers demonstrated low removal efficiencies for protein. Increased removal rates were attained only after operating the flocculation system for 10 to 15 h, albeit turbidity removal was comparatively high across a broad range of flow rates.

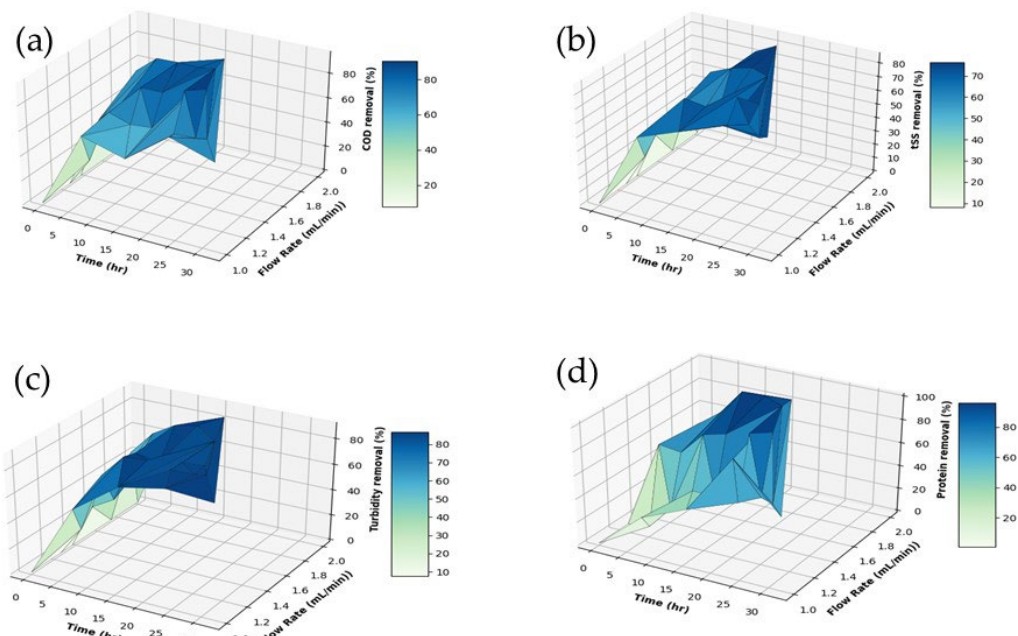

**Figure 8.** Surface plots of (**a**) COD, (**b**) TSS, (**c**) turbidity, and (**d**) protein removal for moulded diffusers.

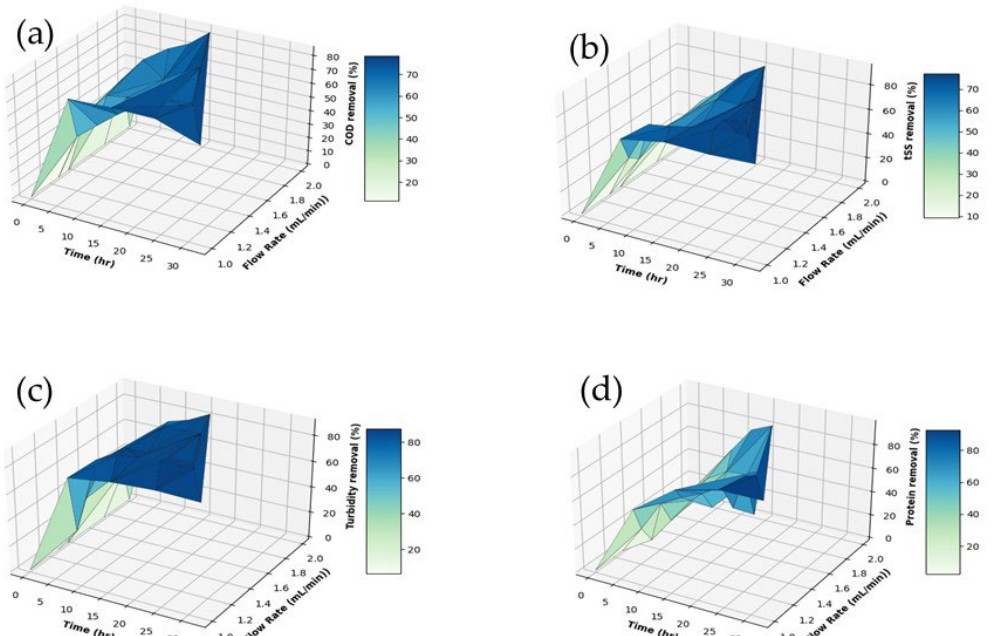

**Figure 9.** Surface plots of (**a**) COD, (**b**) TSS, (**c**) turbidity, and (**d**) protein removal for 3D-printed air diffusers.

*3.5. Overall Bioflocculant Column Flotation (BioCF) System Performance*

The flotation method is a pretreatment technology employed to remove organic matter, different types of suspended solids, oils, and other pollutants from wastewater [42]. The PSW was subjected to physicochemical analysis before the pretreatment process was initiated. Generally, the wastewater exhibited a substantial organic load (COD, TSS, and proteins), which also led to the wastewater's turbidity. The variation in process parameters regarding feed flow rate, bioflocculant form, and diffuser type resulted in varying system performance regarding pollutant removal. The performance of the flotation system was evaluated based on pollutant removal efficiencies for COD, TSS, turbidity, and protein reduction, as illustrated in Figures 10–13.

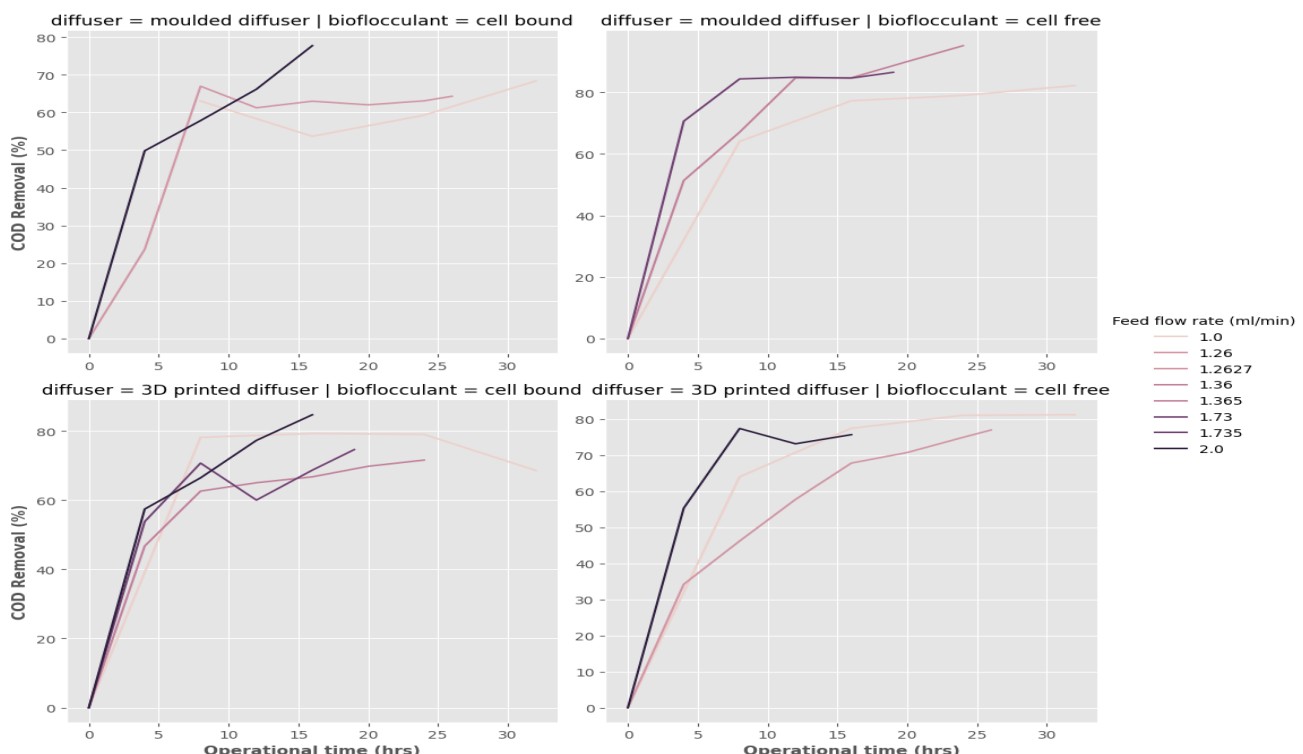

**Figure 10.** Graphical representation of COD removal under various conditions (diffuser type, bioflocculant form, and feed flow rate).

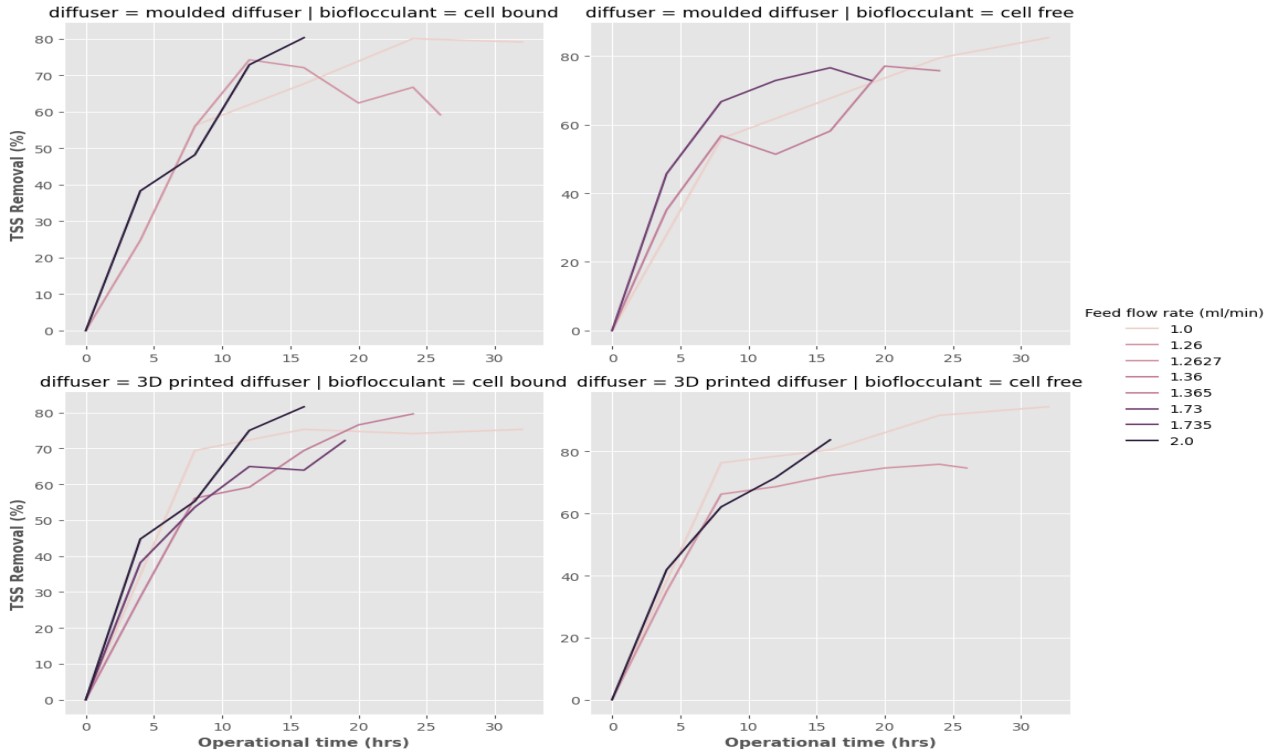

**Figure 11.** Graphical representation of TSS removal under various conditions (diffuser type, bioflocculant form, and feed flow rate).

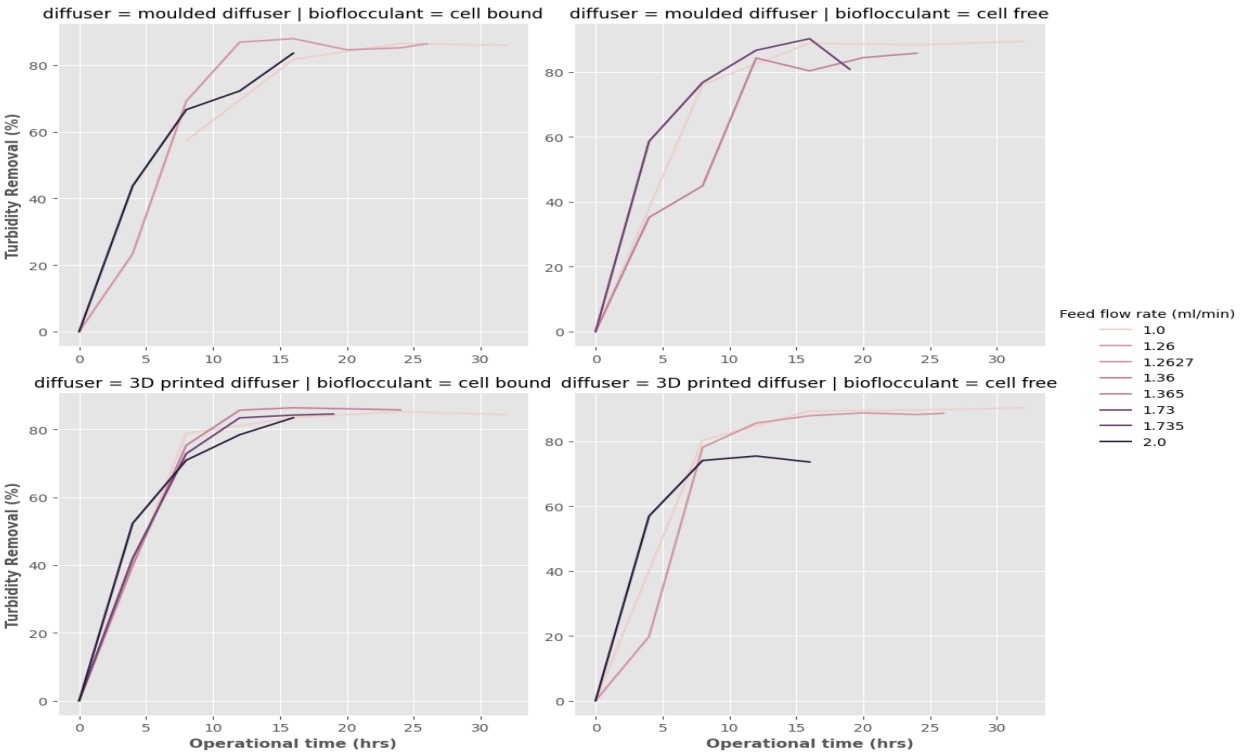

**Figure 12.** Graphical representation of turbidity reduction under various conditions (diffuser type, bioflocculant form, and feed flow rate).

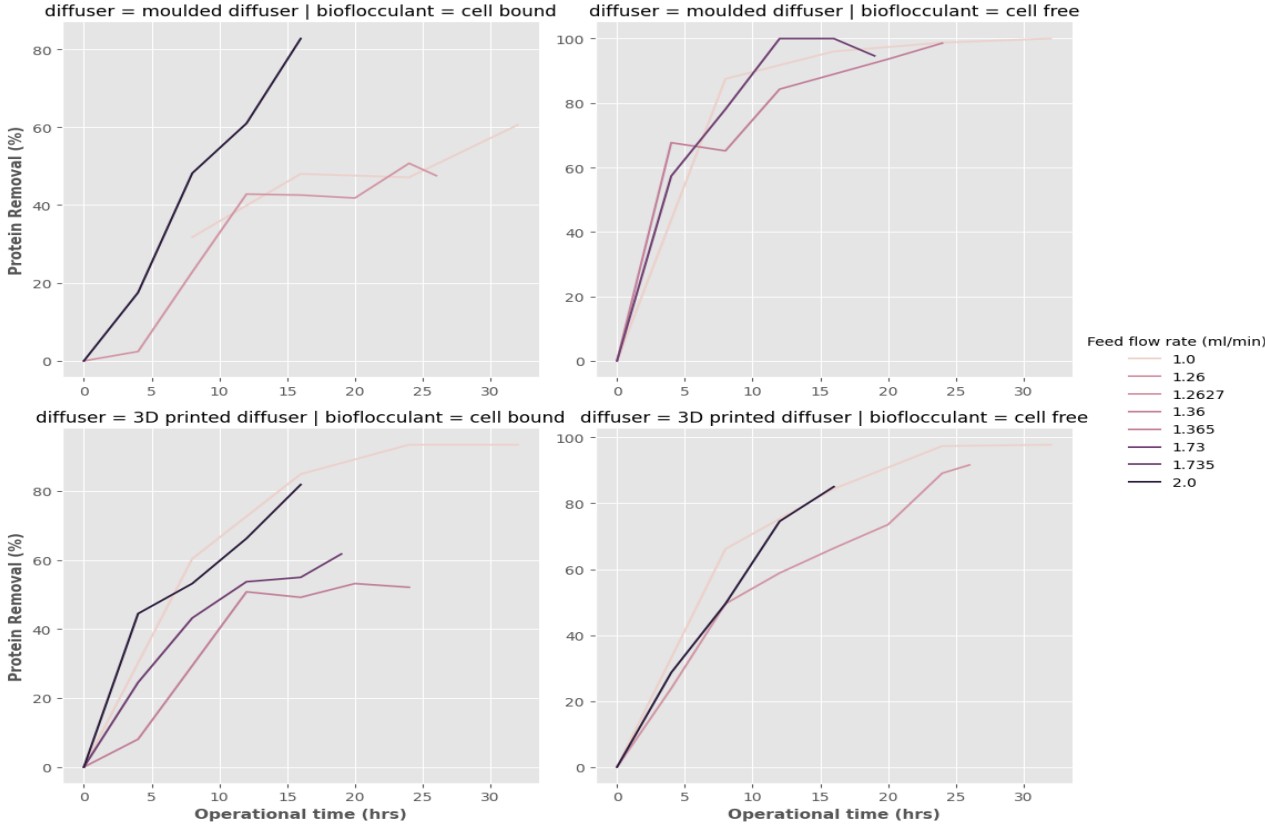

**Figure 13.** Graphical representation of protein reduction under various conditions (diffuser type, bioflocculant form, and feed flow rate).

The COD removal rate is typically used to measure the strength and treatability of wastewater [43], thus, organic matter content in the wastewater [44]. The graphs in Figure 10 depict moulded diffusers' performance with cell-free flocculants, indicating higher COD removal rates, with removal efficiencies of over 80% for all three-feed flow rates. The removal was the least, ranging between 60% and 80% for moulded diffusers with cell-bound bioflocculant. Additionally, the maximum removal of COD was observed at a flow rate of 1.365 mL/min.

Solids such as soft tissue, excrement, feathers, etc. are responsible for high TSS values in PSW [44]. The graph in Figure 11 shows that TSS removal was high for 3D-printed diffusers with cell-free flocculants. It can be noticed that at a flow rate of 2 mL/min, the TSS removal efficiency was just above 80%, as well as at the flow rate of 1 mL/min, where TSS removal rates were the highest except for a combination of cell-bound flocculants with 3D-printed air diffusers.

Turbidity in PSW is also elevated by blood and urine apart from suspended solids. Regarding turbidity reduction (Figure 12), the flotation process proved effective as most results were above an 80% removal efficiency across a range of flow rates assessed.

Figure 13 shows that the system was not too effective in reducing the protein content of the PSW, mainly where cell-bound flocculants were used; an attribute also associated with both 3D-printed and moulded diffusers. This could have been attributed to increased microbial community proliferation as the PSW. However, the protein removal efficiency was relatively high for a flow rate of 1 mL/min, as this would have resulted in increased hydraulic retention time for the system.

Ref. [38] (2008) used a column flotation to treat meat processing wastewater. The authors found that it had acceptable removal efficiencies of organic matter and, thus, recommended that it be used as a cheap alternative to dissolved air-flotation systems, provided the right flocculants are used. Although DAF systems are known for high removal efficiencies, according to most reports, column flotation can achieve such high removal efficiencies at relatively low costs with minimal maintenance, provided that parameters affecting the flotation process are optimized.

Based on the findings, it is evident that the system successfully pretreated PSW. Furthermore, it shows that the selected variables affected the flotation system performance. On average, a 1 mL/min flow rate combination with 3D-printed diffusers and cell-free bioflocculants yielded high pollutant removal. Overall, treatment with cell-free flocculants achieved relatively high removal efficiencies compared to cell-bound flocculants. When cell-bound flocculants were used, it is apparent that protein removal efficiencies were poor and there were lower COD removal rates as opposed to when cell-free flocculants were used. This further confirms that the form of bioflocculant affects other parameters, especially on diffusers, as they will be linked to poor pollutant-removal efficiencies. The form of bioflocculants and type of diffusers affect the performance of a bioflocculant-supported column flotation. The flow rate did not exclusively affect the removal efficiencies as the results varied.

## 4. Conclusions

Removing pollutants from PSW using a bioflocculant-supported column flotation proved an effective pretreatment method. The performance of the system was affected by various parameters, and the following conclusions stand out: the best bioflocculant form was cell-free flocculants, and the best diffuser type was the 3D-printed air diffuser as high removal rates were attained when these variables were employed as compared to cell-bound flocculants and moulded diffusers. There is a need to explore more manufacturing 3D-printed diffusers in terms of improving the diffusers themselves and employing them in different types of wastewaters as the results reflected that they can improve the flotation system performance. It is further recommended that the BioCF system's long-term stability and scalability in an industrial setting be investigated, along with a detailed assessment of



its overall environmental impact, including the lifecycle analysis of the 3D-printed diffusers and bioflocculants.

**Author Contributions:** Conceptualization, M.R.M., S.K.O.N., M.B., N.M. and C.D.; methodology, C.D.; software, M.N.; validation, S.K.O.N. and M.B.; formal analysis, M.R.M.; investigation, M.R.M.; resources, S.K.O.N. and M.B.; data curation, M.R.M. and M.N.; writing—original draft preparation, M.R.M.; writing—review and editing, S.K.O.N., M.B. and B.S.C.; visualization, C.D.; supervision, S.K.O.N., M.B., and B.S.C.; project administration, M.R.M.; funding acquisition, M.B. and S.K.O.N. All authors have read and agreed to the published version of the manuscript.

**Funding:** This research was funded by the National Research Foundation Thuthuka Funding grant number 138173, Cape Peninsula University of Technology South Africa, grant number RK45, and the CPUT Vice Chancellor Achiever's Award. Furthermore, a scholarship for M.R.M. was provided for by Mwalimu Nyerere African Union scholarship scheme.

**Data Availability Statement:** The data presented in this study are available on request from the corresponding author.

**Conflicts of Interest:** Author M. Njoya was employed by the company Atos. The remaining authors declare that the research was conducted in the absence of any commercial or financial relationships that could be construed as a potential conflict of interest. The funders had no role in the design of the study; in the collection, analyses, or interpretation of data; in the writing of the manuscript; or in the decision to publish the results.

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
