# Peer review of "Evaluation of Selected Operating Process Variables for a Bioflocculant Supported Column Flotation System"

_water, doi:10.3390/w16020329_

Round 1

Reviewer 1 Report

Comments and Suggestions for Authors

The study investigates the impact of three variables—diffuser design, bioflocculant form, and flow rate—on the performance of a bioflocculant-supported column flotation (BioCF) system, particularly for treating poultry slaughterhouse wastewater (PSW). It found that 3D-printed air diffusers and cell-free bioflocculants significantly enhance BioCF performance, achieving high pollutant removal efficiencies. It indeed worth publication.

Some points can be improved:

1.     The Introduction part can be improved to be more related to the theme of this manuscript. One of the conclusion of this manuscript is that “the combination of 3D-printed air diffusers and cell-free flocculants resulted in relatively high pollutant removal” However, more background of 3D-printed air diffusers and cell-free flocculants is not included.

2.     The data lack of more comparison with other current method. It need to present that how advanced this methods be by comparing with traditional methods.

3.     In 2.2 Wastewater source, the parameters should be presented at the very beginning. And Table 2 have no units.

4.     The paper does not address the long-term stability and scalability of the BioCF system in an industrial setting.

5.     Although the study emphasizes the ecofriendly nature of the system, a detailed assessment of its overall environmental impact, including the lifecycle analysis of the 3D-printed diffusers and bioflocculants, is missing.

Comments on the Quality of English Language

English is good enough for a scientific publication.

Author Response

Dear Esteemed Reviewer,

We extend our sincere gratitude for taking the time to review our work. Your valuable feedback has been carefully considered, and we are pleased to inform you that we have incorporated the suggested improvements into the submitted manuscript. We believe these revisions have significantly enhanced the overall quality of our work.

Your insightful comments have played a crucial role in refining our research, and we appreciate your dedication to maintaining the scholarly standards of our field. Your expertise has been instrumental in shaping this contribution to the academic community.

If there are any further suggestions or concerns, we remain open to additional feedback to ensure the continual enhancement of our work. We are committed to delivering a manuscript that aligns with the highest standards of academic excellence.

Thank you once again for your time, effort, and expertise.

Warm regards,

Prof Moses Basitere

Reviewer 2 Report

Comments and Suggestions for Authors

This paper is a study to combine 3D-printed air diffusers and microbial flocculants. 3D technology is one of the critical technologies of the future and deals with interesting topics, but this study also only examines one type of 3D diffuser. It is not expected to be able to judge the effectiveness of this technology. Also, from Fig. 4 onward in this paper, the conclusions obtained are the same, and it is unnecessary to include all the figures. It should consist of only the necessary figures; the rest should be attached in an appendix.

1) p.6 Line 200~205: The use of kaolin appears in the results but is not mentioned in the experimental methods section.

2) Tables 1 and 2 should be written in the experimental methods section.

3). The scale on the vertical axis in Figure 3 is not visible.

4). What does 210-340 mg/L and 110-180 mg/L mean? The values do not match the vertical axis of Fig. 3. Please explain carefully. Also, it says "significantly changed," but was this determined by p<0.05? The M&M did not state the criterion. Please specify p<0.05 or. p<0.01, as it was written that statistical treatment was performed.

5) Fig 4: There is no reason to separate b) and c), e) and f), h) and i), k) and l). It should stop taking up extra space.

Author Response

(The authors gave the same response as above.)

Round 2

Reviewer 2 Report

Comments and Suggestions for Authors

The revisions were well done. This paper is acceptable.